# Association of *CYP24A1* Gene rs6127099 (A > T) Polymorphism with Lower Risk to COVID-19 Infection in Kazakhstan

**DOI:** 10.3390/genes14020307

**Published:** 2023-01-24

**Authors:** Antonio Sarría-Santamera, Kymbat Mukhtarova, Ardak Baizhaxynova, Kaznagul Kanatova, Saule Zhumambayeva, Ainur Akilzhanova, Azliyati Azizan

**Affiliations:** 1School of Medicine, Nazarbayev University, Astana 010000, Kazakhstan; 2Department of Propedeutics of Children Disease, Astana Medical University, Astana 010000, Kazakhstan; 3National Laboratory Astana, Nazarbayev University, Astana 010000, Kazakhstan; 4College of Osteopathic Medicine, Touro University Nevada, Henderson, NV 89014, USA

**Keywords:** COVID-19, SARS-CoV-2, vitamin D, immune system, metabolism

## Abstract

In December 2019, SARS-CoV-2 was identified in Wuhan, China. Infection by SARS-CoV-2 causes coronavirus disease 2019 (COVID-19), which is characterized by fever, cough, dyspnea, anosmia, and myalgia in many cases. There are discussions about the association of vitamin D levels with COVID-19 severity. However, views are conflicting. The aim of the study was to examine associations of vitamin D metabolism pathway gene polymorphisms with symptomless COVID-19 susceptibility in Kazakhstan. The case-control study examined the association between asymptomatic COVID-19 and vitamin D metabolism pathway gene polymorphisms in 185 participants, who previously reported not having COVID-19, were PCR negative at the moment of data collection, and were not vaccinated. A dominant mutation in rs6127099 (*CYP24A1*) was found to be protective of asymptomatic COVID-19. Additionally, the G allele of rs731236 TaqI (*VDR*), dominant mutation in rs10877012 (*CYP27B1*), recessive rs1544410 BsmI (*VDR*), and rs7041 (*GC*) are worth consideration since they were statistically significant in bivariate analysis, although their independent effect was not found in the adjusted multivariate logistic regression model.

## 1. Introduction

In December 2019, SARS-CoV-2 was identified in Wuhan, China. Infection by SARS-CoV-2 causes coronavirus disease 2019 (COVID-19), which is characterized by fever, cough, dyspnea, anosmia, and myalgia in many cases [1]. Severity varies from patient to patient, and there is no exact explanation for this phenomenon. Namely, in almost 80% of SARS-CoV-2 infected people, mild, and in 20%, severe symptoms are manifested. In half of the severe cases, fatal acute respiratory distress syndrome (ARDS) develops [2]. SARS-CoV-2 utilizes angiotensin-converting enzyme 2 (ACE2) receptors located in respiratory tracts to enter host cells. Failure to eliminate the virus at early stages by immune response leads to disease progression and potential adverse outcomes as lung inflammation and fibrosis occur [3].

There are significant controversies regarding the role of vitamin D in COVID-19 severity, controversies that are common to various health problems. Low levels of vitamin D have been associated with a higher risk of severe COVID-19 infection [4,5]. However, the benefits of vitamin D supplementation remain controversial: while some studies show a beneficial effect of vitamin D supplementation in COVID-19 severity [6,7,8], others fail to find any associations [9]. 

Nevertheless, the role of vitamin D in immune function is a well-studied topic. Thus, although the situation with supplementation is not clear, there is consistent evidence of the relevance of vitamin D metabolism in the immune system. Vitamin D receptors (VDR) are present in immune cells such as antigen-presenting cells, T and B cells, and monocytes [10,11]. Jeng et al. [12] reported that vitamin D and vitamin D binding proteins (DBP encoded by the GC gene) were critically low in sepsis patients. Sufficient levels of DBP are necessary to recover 25OH vitamin D (25(OH)D3) loss in the urine [12]. 

Variations in vitamin D metabolism-associated human genes are found to be associated with varied susceptibility to diseases. For instance, the VDR gene is in chromosome 12 and has widely-studies polymorphisms at TaqI (rs731236), FokI (rs2228570), BsmI (rs1544410), and ApaI (rs7975232) restriction sites. TaqI, BsmI, and ApaI are in strong linkage-disequilibrium as all are located in the 3′ end of VDR. FokI, in turn, is in the start codon. Thus, all of them functionally affect VDR function [13]. Recently, Zeidan et al. [14] reported an association of FokI polymorphism with COVID-19 susceptibility. 

There are also GC (group-specific component) single nucleotide polymorphisms rs4588 and rs7041 reported to affect the affinity of vitamin D binding proteins to vitamin D metabolites [15]. Also, studies reported an association of rs4588 and rs7041 genotypes with COVID-19 severity [15,16]. *CYP24A1* encodes 25-hydroxyvitamin D3-24-hydroxylase enzyme that degrades 25(OH)D3 into 24.25(OH)2D3. Variant rs6013897 was shown to be associated with inflammatory reactions [17]. 

*CYP27B1* and VDR are expressed in high levels in respiratory epithelial cells, and rs10877012 polymorphism is in the promoter region of the gene [18]. Thus, the polymorphism can affect the expression level of the protein. Polymorphisms in *CYP27A1, CYP2R1,* and *CYP27B1* were reported to be associated with a variety of immune system function-related diseases such as multiple sclerosis, autoimmune thyroid diseases, viral infections, and type 1 diabetes mellitus [19]. Rs1800629 affects the transcription of the *TNF-α* gene. *TNF-α* promoter polymorphism rs1800629 is associated with susceptibility and severity of COVID-19 [20]. 

The aim of the study was to examine associations of vitamin D metabolism pathway genes polymorphisms that previous research has shown to be associated with suspected critical roles of vitamin D in susceptibility to COVID-19 infection in participants who reported not having been infected with COVID-19 in Kazakhstan.

## 2. Materials and Methods

### 2.1. Data Collection

The setting of this study was Olymp Laboratories in the city of Astana, Kazakhstan. From September 2021 to December 2021, conditionally healthy men and women above 18, who claim that they have never had COVID-19, were not vaccinated and were PCR negative at the moment of data collection, and who came for routine blood tests for any indication were invited to participate in this study. During the recruitment process, each participant was provided with a free PCR test, measurement of total IgM/IgG antibodies against SARS-CoV-2, and serum 25(OH)D3 levels measurement. Only participants with negative PCR were included in the study. Cases and controls were separated based on cut-off indexes (COI) provided by the Olymp Laboratories. The COI is determined by comparing samples to the positive control. The COI is derived from the ratio of sample signal vs. positive control signal. Samples with COI ≥ 1.0 of IgM/IgG levels were diagnosed as IgM/IgG positive and taken as asymptomatic cases, whereas those with total antibodies COI < 1.0 were interpreted as negative for IgM/IgG. Questionnaires were also provided by specially trained healthcare workers. Ethical approval was obtained from the Nazarbayev University Institutional Research Ethics Committee (422/11062021). 

### 2.2. Questionnaire

The questionnaire consisted of three main parts:Socio-demographic characteristics;Previous Medical History;The lifestyles of the participants.

Socio-demographic questions included information about age, sex, ethnicity, height, and weight. Participants were asked about their medical history of chronic diseases, such as stroke, cancer, diabetes, asthma, allergy, high blood pressure, high cholesterol, and heart, lung, and kidney-related disorders. Moreover, questions about the absence and presence of COVID-like symptoms during the last six months and BCG vaccination were included in the questionnaire. Questions about participants’ lifestyles were related to their smoking status and alcohol use, and regular sports activities they do. Also, they were asked whether they worked/studied during the pandemic period. 

### 2.3. Genotype Data

Whole blood samples of participants were de-identified and collected in EDTA-containing vacutainer tubes by medical personnel of Olymp laboratories. 

DNA was extracted by use of Wizard Genomic DNA Purification Kit (Promega, Madison, WI, USA) according to the manufacturer’s protocol. Quantitation and quality of DNA were ascertained using a NanoDrop 2000 spectrophotometer (Thermo Fisher Scientific, Wilmington, DE, USA). 

Genotyping was performed using qualitative real-time PCR (Bio-Rad, Hercules, CA, USA) in 384-well plates. Thermal cycling conditions were as follows: polymerase activation at 95 °C for 10 min followed by 40 cycles of denaturation (at 95 °C for 15 s) and annealing extension (at 60 °C for 1 min).

### 2.4. Statistical Analysis

Data cleaning was performed using Microsoft Excel. All statistical analysis was conducted using the Stata 14.2 (Stata Corporation, College Station, TX, USA) statistical program and SNPStats online tool based on R [21].

Basic descriptive statistics, such as frequencies and mean values, were generated. To assess association with the outcome variable, Fisher’s exact test was used for categorical independent variables, and the Wilcoxon Rank Sum test was used for continuous independent variables. To estimate the strength of the association between polymorphisms and COVID-19, multivariate logistic regression analysis was performed. Demographic covariates were included in the adjusted model to adjust for their possible confounding effect on the outcome variable. The odds ratio (OR) and 95% confidence interval (CI) were calculated. Linkage disequilibrium and haplotype analysis were conducted.

Participants were divided into two groups: cases (with positive SARS-CoV2 antibodies indicating previous infections (COI ≥ 1), and which may be considered asymptomatic cases) and controls (with negative SARS-CoV2 antibodies (COI < 1) and PCR test). 

All statistical tests were two-sided. Following the Bonferroni correction or multiple comparisons, *p* < 0.0046 was taken as significant for vitamin D metabolism pathway genetic associations analysis. A significance level (α) equal to 0.05 was chosen for descriptive statistics.

The Hardy-Weinberg equilibrium test and bivariate statistics for the different inheritance patterns were conducted as well.

## 3. Results

### 3.1. Demographic Data

One hundred eighty-five participants were recruited for this study, but complete data for analysis was only available for 180. The sociodemographic data of study subjects are summarized in Table 1. 64.9% of cases and 56.5% of controls were females (*p* > 0.05). Age, BMI, and serum vitamin D levels were comparable in cases and controls (*p* > 0.05).

### 3.2. Association Study

Overall, 12 SNPs were genotyped. The analysis showed that rs7975232 ApaI (*VDR*) was not in Hardy–Weinberg equilibrium (*p* < 0.05) (Table 2). 

Minor allele frequencies of 11 SNPs were compared (Figure 1). Rs731236 TaqI (*VDR*), rs1544410 BsmI (*VDR*), and rs6127099 (*CYP24A1*) had statistically significant differences in MAF (minor allele frequency) (*p* ≤ 0.05).

Associations between SNPs and genotypes (bivariate analysis) under additive, dominant and recessive models are summarized in Table 3. A recessive inheritance pattern is when two copies of a risk allele are necessary to cause an effect. In turn, the dominant mode of inheritance depicts situations when it is enough to have at least one copy of a mutated allele to cause an effect. In the additive inheritance pattern, the effect increases with each copy of the mutated allele.

From this, only rs6127099 (CYP24A1) was statistically significant based on both Bonferroni corrected and baseline *p*-values under the dominant mode of inheritance (*p* = 0.004). Namely, people with at least one copy of the mutant T allele in rs6127099 have 84% to 29% less symptomless COVID-19 than AA-genotyped participants. In the allelic model, the T allele was found to be associated with OR = 0.46 (0.28–0.76) of asymptomatic COVID-19.

Other SNPs were found to be significant at *p* = 0.05 level. Namely, VDR gene polymorphism at rs1544410 (BsmI) site revealed that people with CC genotype have OR = 0.44 95% CI (0.21–0.91) asymptomatic COVID-19 than those with TT and TC genotypes under the recessive mode of inheritance. Similarly, T > C allelic shift resulted in 0.49 (0.25–0.94) times the odds of asymptomatic COVID-19 vs. controls. Similarly, recessive rs7041 in the GC gene was statistically significant with OR = 0.37 (0.15–0.93) times the chance of COVID-19 in CC than in AA or AC carriers. The dominant mode of inheritance of rs10877012 polymorphism of CYP27B1 gene reported increased association of the T allele with asymptomatic COVID-19 with OR = 2.51 (1.05–5.99).

Collinear variables were not found. To identify confounders, logistic regression analysis was applied. No statistically significant association was found between demographic, clinical, and behavioral variables and asymptomatic COVD–19.

Overall, the odds of asymptomatic COVID-19 are 0.32 (0.15–0.68) times for CYP24A1 (rs6127099) in AT + TT genotyped participants in comparison with AA-genotyped people adjusted for age, male gender, and Kazakh ethnicity. 

The allele-based model states that there are decreased odds of asymptomatic COVID-19 in rs6127099 T vs. A OR = 0.44 (0.26–0.74), adjusted for age, male gender, and Kazakh ethnicity. Stratifications based on gender (male vs. female) and age categories (less than 40 (<40); 40 and 59.9 (>=40 and <60); and 60 + (60<=) can be found in Appendix A.

### 3.3. Association of VDR, GC, and CYP24A1 Haplotypes with Asymptomatic COVID-19

Linkage disequilibrium showed a certain level of deviation from expected genotype frequencies in our sampling (D’ ≠ 0). Especially, it was prominent in combinations of rs731236 and rs1544410 of VDR (*p* = 0.000); rs7041 and rs4588 of GC (*p* = 0.000); and rs6013897 and rs6127099 of CYP24A1 (*p* = 0.000). This points to a possible mechanism of co-segregation at those sites.

From haplotype analysis (Table 4), the GGT haplotype of VDR (block 3) was found to be statistically significantly associated with asymptomatic COVID-19 susceptibility OR = 3.12 (1.07–9.10). Block 11 (G allele of rs731236 (TaqI) and T allele of rs1544410 (BsmI)) of VDR was found to be associated (*p* = 0.028) with increased odds of asymptomatic COVID-19 95% CI is between 1.1 and 4.61. Similarly, the G allele of FokI and the T allele of BsmI are associated with 2.90 (1.06–7.91) times of asymptomatic COVID odds vs. G and C alleles, respectively. Nevertheless, *p*-values here are above the adjusted *p*-value and thus can serve as a baseline for further research, but it is not conclusive in our sampling. In contrast, a statistically significant association (*p* < 0.0046) was identified in CYP24A1 haplotype TT (block 2). There is a decrease in odds ratio, OR = 0.37 (0.19–0.73), of asymptomatic COVID-19 in participants that confer the T allele in rs6013897 and the T allele in rs6127099 vs. TA haplotype. 

## 4. Discussion

The case-control study examined the association between the presence of antibodies against COVID-19 and vitamin D metabolism pathway gene polymorphisms in 180 participants who previously reported not having COVID-19, who were PCR negative when data was collected, and were not vaccinated in Kazakhstan.

This study showed that a dominant mutation in rs6127099 (*CYP24A1*) appears to be associated with negative anti-COVID-19 antibodies. Additionally, potentially protective recessive rs1544410 BsmI of VDR gene and rs7041 of GC gene, and G allele of rs731236 TaqI (*VDR*) and dominant mutation in rs10877012 (*CYP27B1*) that can potentially increase the susceptibility of asymptomatic COVID-19 are worth considering since they were statistically significant in bivariate analysis.

A relevant finding of this work is the high number of asymptomatic COVID-19 cases identified. This high number of asymptomatic cases is a public health concern since those cases have the same risk of transmitting the infection [22]. Asymptomatic cases represent a substantial limitation to infection control measures. Moreover, the high proportion of asymptomatic cases may imply that the actual number of infections may be much higher than reported by public health authorities [23]. Both groups reported a high but not statistically significant proportion of COVID-like symptoms.

Nevertheless, at the individual patient level having asymptomatic COVID-19 without serious clinical complications can be somewhat more beneficial than having severe COVID-19-related symptoms. This can be explained by host genetic differences along with well-known advanced age, male sex, and chronic diseases [24]. 

Finally, although the results from this study reflect the lack of association of vitamin D levels with the risk of having or not having a previous COVID-19 infection, our work reveals the association of various genetic factors related to the metabolic pathways of vitamin D with the risk of asymptomatic infection.

Vitamin D has a significant role in the adaptive immune response. Namely, the adaptive immune system includes major players such as dendritic cells (DC) and macrophages that are essential for antigen presentation. They, in turn, activate antigen-recognizing T and B lymphocytes. 1,25(OH)2D3 is known to decrease the maturation of DCs. Furthermore, 1,25(OH)2D3 suppresses Th1 and Th17 development caused by reduced production of IL–12 and IL–23, IL–6, respectively. Noticeably, Th1 cells produce IFN–γ, IL–2, and Th17 cells produce IL–17. In turn, IFN–γ deficiency leads to the prevention of T-lymphocyte recruitment, and IL–2 deficiency leads to disturbed T-lymphocyte proliferation. Suppression of IL–12 leads to the development of Th2 cells that causes an increase of IL–4, IL–5, and IL–10 that, again, suppress Th1 development. Thus, the balance shifts towards more Th2 phenotype [11]. This means that the body avoids a prolonged inflammatory response and its damaging effects since it is known that there is increased expression of pro-inflammatory cytokines IL–1β, IL–6, TNF, IL–12, IFN–β, IFN–γ, IL–17 in COVID-19 [25]. Failing to shift from pro- to anti-inflammatory is linked with cytokine storms commonly observed in severe SARS-CoV-2 infection.

Innate immunity is the first line of defense against any infection. In the case of COVID-19, innate immunity detects SARS-CoV-2 through pattern-recognition receptors (TLR1, TLR4, and TLR6) and activates downstream cascades to initiate viral clearance [25]. Vitamin D is known to decrease DC maturation, enhance macrophage differentiation, enhance bacterial killing, lowering cytokine levels and antigen presentation [26]. Once detected by Toll-like receptors (TLR), pathogen invasion induces expression of *CYP27B1* and VDR, favoring the production of cathelicidin, which acts against bacteria, viruses, and fungi by primarily destabilizing microbial membranes from macrophages and neutrophils [10,12]. In addition, IFN–γ and IL–4 are also known to enhance the expression of *CYP27B1*.

CYP24A1 is responsible for the inactivation of active metabolites of vitamin D. Interestingly, *CYP24A1* rs6127099 polymorphism is also known to be associated with elevated parathyroid hormone concentrations [27,28], which in turn leads to elevated calcium levels. The presence of *CYP24A1* mutations has been linked with increased sensitivity to vitamin D [29]. Thus, these findings bring room for further investigations of the role of calcium in SARS-CoV-2 infection. Lower calcium levels have been reported to be associated with COVID-19, and its severity [30,31,32] and COVID-19 infection has been suggested to occur in the context of marked hypovitaminosis D not adequately compensated by secondary hyperparathyroidism [33].

The kidney, acting as an endocrine gland, converts 25(OH)D3 by the action of the enzyme 1α-hydroxylase (*CYP27B1*) to the active hormonal form 1α, 25-dihydroxyvitamin D [1,25(OH)2D], known as calcitriol. *CYP27B1* is expressed in macrophages and dendritic T and B cells and is known to affect calcitriol levels [34]. Calcitriol then binds to VDR, a member of the nuclear receptor family, which is a receptor specific to vitamin D through which vitamin D exerts its function. VDR binds to the active form of intracellular vitamin D to interact with the nuclei of the target cells. Calcitriol signaling is crucial in bone metabolism as it is involved in calcium absorption, parathormone secretion, and, therefore, bone resorption and cellular differentiation, but it also has immunological functions as well as different functions in different body organs. VDR has many polymorphisms. TaqI is one of those VDR gene polymorphisms. Those polymorphisms have been associated with several health problems and may modulate vitamin D functions [35].

TaqI polymorphisms have also been identified to be associated with a higher risk of COVID-19 infection [36,37,38,39], showing rs731236 as significantly associated with a severe type of infection and association with ICU admission. Two studies in Iran did not find an association with TaqI, but they only included hospitalized cases [40,41].

It is complicated to compare our findings with other studies because, to date and to the best of our knowledge, there has been no genetic study conducted with participants who reported not having COVID-19 before. However, there are studies with conflicting findings about the role of serum vitamin D in COVID-19 infection. For instance, Bouillon and colleagues [42] reported that supplementation of vitamin D–replete individuals (baseline serum 25-OH vitamin D > 50 nmol/L) does not provide demonstrable health benefits. In contrast, Jain et al. [43] found that vitamin D level was significantly low in severe COVID-19 patients compared to asymptomatic COVID-19. 

A possible hypothesis that explains why vitamin D deficiency is related to a defective immune response and, consequently, to higher mortality, while supplementation with vitamin D does not provide consistent benefit, is the existence of alterations in the complex activation and functioning mechanisms of vitamin D.

Keep in mind that the discrepancies between the higher risk associated with low vitamin D levels and lack of benefit vitamin D supplementation are not exclusive to SARS-CoV2 infection but have been identified in numerous health problems previously.

Alterations in adaptive immunity and vitamin D status can affect the prognosis of COVID-19 by affecting bone metabolism. Under inflammatory conditions, cytokines, such as tumor necrosis factor (TNF), IL-6, and IL-1, can upregulate osteoclastogenesis and inhibit osteoblast activities. TNF is a key factor in bone loss and might synergize with the receptor activator of nuclear factor kappa-B ligand (RANKL) to induce osteoclastic bone resorption. Activated T and B cells serve as major sources of RANKL and TNF in inflammatory states [44].

The present data suggest that vitamin D metabolism may be associated with COVID-19 infection. However, in our study, 25(OH)D3 serum levels were not associated with differences in the presence of SARS-CoV-2 antibodies. The reasons for these discrepancies remain unclear, but it is well-known that 25(OH)D3 serum levels correlate poorly with calcitriol serum concentrations, and 25(OH)D3 serum levels are therefore not a suitable marker for bioactive vitamin D or vitamin D receptor signaling [45]. 

Thus, the lack of an association between 25(OH)D3 serum levels and antibodies may simply reflect the limited biological relevance of 25(OH)D3 serum levels. Unfortunately, there are no reliable methods to quantify serum levels of the bioactive vitamin D metabolite calcitriol, and most clinical trials assessing the vitamin D status of patients focus on the calcitriol precursor 25(OH)D3.

The study has several limitations. Firstly, the low sample size limits the power of this study to detect significant differences, meaning that results obtained here may be subject to type I error. However, it should not be forgotten that hypothesis testing for the statistical significance of any effect depends collectively on three intertwined parameters: the size of the effect, the sample size, and the variability present in the sample data. Although during the recruitment, we aimed to include as many participants as possible, there was not possible to increase the sample size. 

Another limitation is that we cannot determine the accuracy of participants’ indications of not having been previously diagnosed with COVID-19. A related limitation is that we cannot clearly elucidate the existence of differences between cases in controls in exposure to COVID-19.

Also, to mention that our serological analysis did not differentiate IgM and IgG to detect early or later infections, but IgM–IgG combined antibody detection is a more reliable method, with greater specificity and sensitivity compared with single IgM or IgG tests [44]. Another relevant issue is the lack of a control group with participants who had clinically manifested COVID-19. The ethnic diversity and characteristics of the Kazakhstani population analyzed also make it complex to extrapolate results to other settings and populations. These findings may be valid for the specific variants which circulated in Kazakhstan before the study started (September 2021). 

Our findings further elucidate genetic susceptibility to COVID-19 infections and may lead to the design of personalized preventive measures to decrease morbidity and mortality due to the SARS-CoV-2 pandemic [18]. In future studies that analyze the role of vitamin D in susceptibility to SARS-CoV-2 infection and other conditions, vitamin D levels have to be investigated in conjunction with the participants’ genetic profiles to further understand the possible protective effect role of vitamin D.

## 5. Conclusions

The study examined the role of socio-demographic, clinical, and individual genetic characteristics of the vitamin D metabolism pathway of unvaccinated, SARS-CoV-2 PCR-negative, and self-claimed symptomless people in asymptomatic COVID-19 predisposition. In this study, we demonstrated that genetic variances in the vitamin D pathway might modulate susceptibility to and severity of COVID-19 infection. All in all, genetic associations with a dominant mutation in rs6127099 (CYP24A1) showed a reduced frequency with previous COVID-19 infection. However, the low sample size may represent that this study has limited power to detect the true association between genotypes and the presence of COVID-19 antibodies.

## Figures and Tables

**Figure 1 genes-14-00307-f001:**
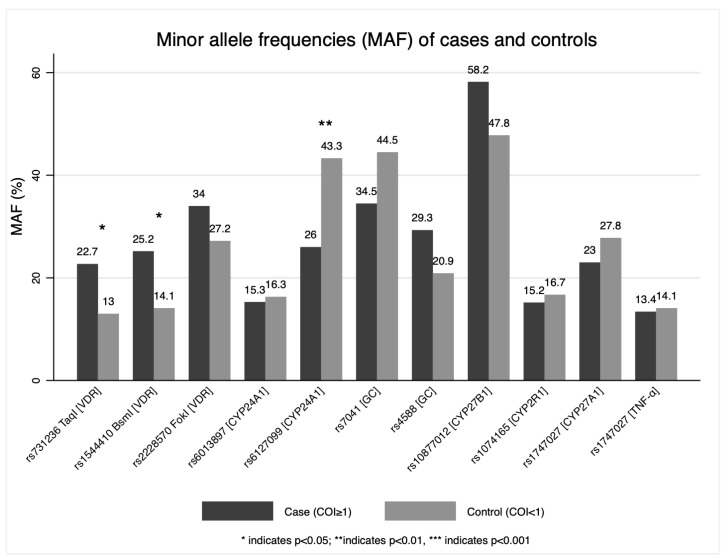
Minor allele frequencies (MAF) of genes of the vitamin D metabolic pathway.

**Table 1 genes-14-00307-t001:** Demographic and clinical characteristics of people with asymptomatic COVID-19 (COI ≥ 1) and healthy controls (COI < 1).

Variables	Case (134)		Control (46)		Total (180)		*p*-Value
**Age**	41.49 ± 13.09		43.09 ± 14.41		41.89 ± 13.26		0.639
**Gender**							0.377
**Male**	47	35.1%	20	43.5%	67	37.2%	
**Female**	87	64.9%	26	56.5%	113	62.8%	
**BMI**	24.5 ± 4.7		24.27 ± 3.86		24.41 ± 4.44		0.868
**BCG scar size**	5.7 ± 3.04		5.24 ± 2.71		5.58 ± 2.91		0.504
**Vitamin D**	28.66 ± 15.3		28.34 ± 12.77		28.58 ± 14.66		0.950
**Ethnicity**							0.580
**Kazakh**	121	90.3%	40	87%	161	89.4%	
**Other**	13	9.7%	6	13%	19	10.6%	
**Heart disorder**	10	7.5%	7	15.2%	17	9.4%	0.145
**Stroke**	1	0.8%	0	0.0%	1	0.6%	1.000
**Cancer**	3	2.2%	3	6.5%	6	3.3%	0.175
**Diabetes**	6	4.5%	3	6.5%	9	95%	0.695
**Asthma**	4	3%	0	0.0%	4	2.2%	0.574
**Hypertension**	18	13.4%	9	19.6%	27	15%	0.342
**High cholesterol**	18	13.4%	4	8.7%	22	12.2%	0.602
**Chronic kidney disease**	18	13.4%	9	19.6%	27	15%	0.342
**Lung disorder**	13	9.7%	2	4.3%	15	8.3%	0.361
**Allergy**	54	40.3%	20	43.5%	74	41.1%	0.731
**Alcohol**	65	48.9%	22	47.8%	87	48.6%	1.000
**Sport**	61	45.5%	25	54.3%	86	47.8%	0.311
**Self–reported COVID-like symptoms**	87	64.9%	30	65.2%	117	65%	1.000
**Employment**	91	68.4%	30	66.7%	121	58%	0.855
**Smoke**	50	37.6%	12	26.1%	62	34.6%	0.208

“COI” means “cut–off interval.” COI < 1 was interpreted as negative for IgM/IgG, and COI ≥ 1 was positive for IgM/IgG.

**Table 2 genes-14-00307-t002:** Genes of the vitamin D metabolic pathway and their SNPs selected for this study.

Gene	Name	SNP	Function	MAF 1000G	HWE *p*-Values
** *VDR* **	Vitamin D receptor	rs731236 (TaqI)	Intracellular hormone receptor that specifically binds 1,25-dihydroxyvitamin D3 (1,25(OH)2D3) and mediates its effects	0.276558	1.00
		rs1544410 (BsmI)		0.295927	0.57
		rs2228570 (FokI)		0.328474	0.71
		rs7975232 (ApaI)		0.484625	0.002 ¶
** *CYP27B1* **	1–α–hydroxylase	rs10877012	Hydroxylation of 25(OH)D3 into 1,25(OH)2D3	0.348642	0.76
** *CYP24A1* **	24–hydroxylase	rs6013897	Mitochondrial enzyme responsible for inactivating vitamin D metabolites. Parathyroid hormone (PTH) concentration, catabolic enzyme for 1,25(OH)2D3 and 25(OH)D3	0.26238	1.00
		rs6127099		0.330072	0.54
** *GC* **	Vitamin D binding protein	rs7041	Binding, solubilization, and transport of vitamin D and its metabolites	0.381589	0.76
		rs4588		0.207867	0.39
** *CYP2R1* **	Vitamin D 25–hydroxylase	rs1074165	The synthesis of bioactive vitamin D (vitamin D3→25D) (at C25 position).	0.122404	0.33
** *CYP27A1* **	Sterol 27–hydroxylase	rs17470271	The synthesis of bioactive vitamin D (vitamin D3 →25D) (at C24 and C27 positions).	0.252995	0.69
** *TNF-α* **	Tumor necrosis factor α	rs1800629	Potent inducer of CYP27B1	0.090256	0.66

¶ denotes *p* < 0.05, Not in Hardy–Weinberg equilibrium.

**Table 3 genes-14-00307-t003:** Association of candidate SNPs with symptomless COVID-19. Genotype frequencies and inheritance patterns of selected SNPs.

Model	SNP [Gene]	Case/Control	OR (95% CI)	*p*-Value	AOR (95% CI)	*p*-Value
	**rs731236 TaqI [*VDR*]**					
Additive	AA	74/34	Reference		Reference	
	AG	50/12	1.91 (0.9–4.05)	0.089	1.97 (0.91–4.27)	0.085
	GG	4/0	1		1	
Dominant	AA	74/34	Reference		Reference	
	AG + GG	54/12	2.07 (0.98–4.36)	0.056	2.13 (0.99–4.6)	0.053
Recessive	AA + AG	124/46	Reference		Reference	
	GG	4/0	1		1	
Allelic	A	198/80	Reference		Reference	
	G	58/12	1.95 (1.00–3.83)	0.051	1.98 (1.00–3.92)	0.05
	**rs1544410 BsmI [*VDR*]**					
Additive	TT	4/0	Reference		Reference	
	TC	57/13	2.13 (1.02–4.42)	0.043 ¶	2.23 (1.04–4.79)	0.04 ¶
	CC	68/33	1		1	
Dominant	TT	4/0	Reference		Reference	
	TC + CC	125/46	1		1	
Recessive	TT + TC	61/13	Reference		Reference	
	CC	68/33	0.44 (0.21–0.91)	0.027 ¶	0.42 (0.2–0.9)	0.025 ¶
Allelic	T	65/13	Reference		Reference	
	C	193/79	0.49 (0.25–0.94)	0.031 ¶	0.48 (0.25–0.94)	0.031 ¶
	**rs2228570 FokI [*VDR*]**					
Additive	AA	14/4	Reference		Reference	
	AG	61/17	1.02 (0.3–3.52)	0.968	1 (0.29–3.51)	0.995
	GG	56/25	0.64 (0.19–2.14)	0.469	0.6 (0.17–2.03)	0.409
Dominant	AA	14/4	Reference		Reference	
	AG + GG	117/42	0.8 (0.25–2.55)	0.701	0.76 (0.23–2.48)	0.652
Recessive	AA + AG	75/21	Reference		Reference	
	GG	56/25	0.63 (0.32–1.23)	0.176	0.59 (0.3–1.18)	0.137
Allelic	A	89/25	Reference		Reference	
	G	173/67	0.73 (0.43–1.23)	0.231	0.7 (0.41–1.19)	0.186
	**rs6013897 [*CYP24A1*]**					
Additive	AA	3/1	Reference		Reference	
	AT	34/13	0.87 (0.08–9.15)	0.909	0.83 (0.08–8.99)	0.875
	TT	94/32	0.98 (0.1–9.75)	0.986	0.96 (0.09–9.83)	0.972
Dominant	AA	3/1	Reference		Reference	
	AT + TT	128/45	0.95 (0.1–9.35)	0.964	0.92 (0.09–9.37)	0.946
Recessive	AA + AT	37/14	Reference		Reference	
	TT	94/32	1.11(0.53–2.32)	0.778	1.14 (0.55–2.4)	0.723
Allelic	A	40/15	Reference		Reference	
	T	222/77	1.08 (0.57–2.07)	0.813	1.1 (0.57–2.11)	0.769
	**rs6127099 [*CYP24A1*]**					
Additive	AA	68/13	Reference		Reference	
	AT	49/25	0.37 (0.17–0.8)	0.012 ¶	0.35 (0.16–0.76)	0.0092 ¶
	TT	8/7	0.22 (0.07–0.71)	0.011 ¶	0.2 (0.06–0.66)	0.0092 ¶
Dominant	AA	68/13	Reference		Reference	
	AT + TT	57/32	0.34 (0.16–0.71)	0.004 *	0.32 (0.15–0.68)	0.003 *
Recessive	AA + AT	117/38	Reference		Reference	
	TT	8/7	0.37 (0.13–1.09)	0.072	0.36 (0.12–1.07)	0.067
Allelic	A	185/51	Reference		Reference	
	T	65/39	0.46 (0.28–0.76)	0.002 **	0.44 (0.26–0.74)	0.002 **
	**rs7041 [*GC*]**					
Additive	AA	52/15	Reference		Reference	
	AC	65/21	0.89 (0.42–1.9)	0.769	0.92 (0.43–1.99)	0.838
	CC	12/10	0.34 (0.13–0.96)	0.041 ¶	0.33 (0.12–0.92)	0.033 ¶
Dominant	AA	52/15	Reference		Reference	
	AC + CC	77/31	0.72 (0.35–1.46)	0.357	0.72 (0.35–1.48)	0.374
Recessive	AA + AC	117/36	Reference		Reference	
	CC	12/10	0.37 (0.15–0.93)	0.034 ¶	0.34 (0.13–0.87)	0.025 ¶
Allelic	A	169/51	Reference		Reference	
	C	89/41	0.66 (0.4–1.06)	0.087	0.65 (0.4–1.06)	0.083
	**rs4588 [*GC*]**					
Additive	GG	58/26	Reference		Reference	
	GT	55/16	1.54 (0.75–3.18)	0.242	1.61 (0.77–3.36)	0.206
	TT	8/1	3.59 (0.43–30.17)	0.240	4.14 (0.48–36)	0.197
Dominant	GG	58/26	Reference		Reference	
	GT + TT	63/17	1.66 (0.82–3.37)	0.160	1.75 (0.85–3.59)	0.130
Recessive	GG + GT	113/42	Reference		Reference	
	TT	8/1	2.97 (0.36–24.5)	0.311	3.32 (0.39–28.03)	0.270
Allelic	G	171/68	Reference		Reference	
	T	71/18	1.57 (0.87–2.83)	0.134	1.64 (0.9–2.97)	0.105
	**rs10877012 [*CYP27B1*]**					
Additive	GG	15/11	Reference		Reference	
	GT	77/24	2.35 (0.95–5.8)	0.063	2.57 (1.02–6.49)	0.046 ¶
	TT	36/9	2.93 (1.01–8.53)	0.048 ¶	3.13 (1.05–9.35)	0.041 ¶
Dominant	GG	15/11	Reference		Reference	
	GT + TT	113/33	2.51 (1.05–5.99)	0.038 ¶	2.72 (1.11–6.65)	0.028 ¶
Recessive	GG + GT	92/35	Reference		Reference	
	TT	36/9	1.52 (0.67–3.48)	0.320	1.51 (0.65–3.52)	0.338
Allelic	G	107/46	Reference		Reference	
	T	149/42	1.53 (0.94–2.48)	0.089	1.54 (0.94–2.53)	0.084
	**rs1074165 [*CYP2R1*]**					
Additive	GG	89/30	Reference		Reference	
	GA	34/15	0.76 (0.37–1.59)	0.473	0.79 (0.37–1.67)	0.535
	AA	2/0	1		1	
Dominant	GG	89/30	Reference		Reference	
	GA + AA	36/15	0.81 (0.39–1.68)	0.570	0.84 (0.4–1.76)	0.641
Recessive	GG + GA	123/45	Reference		Reference	
	AA	2/0	1		1	
Allelic	G	212/75	Reference		Reference	
	A	38/15	0.9 (0.47–1.72)	0.742	0.92 (0.48–1.78)	0.809
	**rs1747027 [*CYP27A1*]**					
Additive	AA	70/18	Reference		Reference	
	AT	34/16	0.55 (0.25–1.2)	0.133	0.51 (0.22–1.17)	0.111
	TT	9/2	1.16 (0.23–5.83)	0.860	0.89 (0.17–4.68)	0.890
Dominant	AA	70/18	Reference		Reference	
	AT + TT	43/18	0.61 (0.29–1.31)	0.206	0.55 (0.25–1.23)	0.146
Recessive	AA + AT	104/34	Reference		Reference	
	TT	9/2	1.47 (0.3–7.14)	0.632	1.2 (0.24–6.00)	0.826
Allelic	A	174/52	Reference		Reference	
	T	52/20	0.78 (0.43–1.42)	0.411	0.71 (0.38–1.32)	0.280
	**rs1800629 [*TNF-α*]**					
Additive	AA	2/2	Reference		Reference	
	AG	30/9	3.33 (0.41–27.13)	0.260	2.95 (0.33–26.5)	0.333
	GG	95/35	2.71 (0.37–20.01)	0.327	2.43 (0.31–19.26)	0.399
Dominant	AA	2/2	Reference		Reference	
	AG + GG	125/44	2.84 (0.39–20.78)	0.304	2.5 (0.32–19.75)	0.385
Recessive	AA + AG	32/11	Reference		Reference	
	GG	95/35	0.93 (0.42–2.05)	0.863	0.94 (0.43–2.08)	0.880
Allelic	A	34/13	Reference		Reference	
	G	220/79	1.06 (0.53–2.12)	0.858	1.05 (0.53–2.11)	0.885

* *p* < 0.0046; ** *p* < 0.0023; ¶ *p* < 0.05; ‘Reference’ indicates OR = 1; ‘UOR’ indicates Unadjusted OR; ‘AOR’ indicates Adjusted OR. The Adjusted logistic model contains age, gender, and Kazakh ethnicity in addition to genotype/allele data.

**Table 4 genes-14-00307-t004:** VDR, GC, and CYP24A1 haplotypes association with asymptomatic COVID-19.

	Haplotype Analysis								LD Analysis	
Gene	Block	SNP1	SNP2	SNP3	Control	Case	Or (95% CI)	*p*-Value	D´	LD *p*-Value
** *VDR* **		**rs731236 (TaqI)**	**rs2228570 (FokI)**	**rs1544410 (BsmI)**					-	-
	1	A	G	C	0.6429	0.4764	Reference			
	2	A	A	C	0.2158	0.2719	1.70 (0.86–3.34)	0.13		
	3	G	G	T	0.0745	0.1603	3.12 (1.07–9.10)	0.039 ¶		
	4	G	A	T	0.0559	0.0678	1.92 (0.53–6.86)	0.32		
	5	A	G	T	0.0109	0.0237	3.38 (0.39–29.65)	0.27		
	6	A	G	-	0.6542	0.5045	Reference		0.0195	0.7917
	7	A	A	-	0.2153	0.2689	1.63 (0.83–3.18)	0.16		
	8	G	G	-	0.074	0.1558	2.82 (0.98–8.12)	0.056		
	9	G	A	-	0.0564	0.0708	1.84 (0.52–6.45)	0.34		
	10	A	-	C	0.8587	0.7481	Reference		0.9997	0.0000
	11	G	-	T	0.1304	0.2283	2.25 (1.10–4.61)	0.028 ¶		
	12	A	-	T	0.0109	0.0236	3.01 (0.35–25.97)	0.32		
	13	-	G	C	0.6386	0.4807	Reference		0.0524	0.7169
	14	-	A	C	0.2201	0.2675	1.61 (0.82–3.19)	0.17		
	15	-	G	T	0.0896	0.1796	2.90 (1.06–7.91)	0.039 ¶		
	16	-	A	T	0.0517	0.0722	2.20 (0.58–8.38)	0.25		
** *GC* **		**rs7041**	**rs4588**							
	1	C	G	-	0.4457	0.345	Reference		0.9992	0.0000
	2	A	G	-	0.3465	0.3625	1.38 (0.78–2.42)	0.27		
	3	A	T	-	0.2079	0.2925	1.94 (0.99–3.79)	0.055		
** *CYP24A1* **		**rs6013897**	**rs6127099**							
	1	T	A	-	0.5464	0.694	Reference		0.6472	0.0000
	2	T	T	-	0.2906	0.154	0.37 (0.19–0.73)	0.0045 *		
	3	A	T	-	0.1469	0.1078	0.58 (0.27–1.25)	0.17		
	4	A	A	-	0.0162	0.0442	2.01 (0.24–17.03)	0.52		

* *p* < 0.0045; ¶ *p* < 0.05; ‘Reference’ indicates OR = 1.

## Data Availability

Because of ethical reasons data analyzed in this study is not available.

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
