# Peer review of "Association of CYP24A1 Gene rs6127099 (A > T) Polymorphism with Lower Risk to COVID-19 Infection in Kazakhstan"

_genes, 2023, doi:10.3390/genes14020307_

Round 1

Reviewer 1 Report

My main comcern about the statistical methodoloy is that authors should have applied (or at least discuss) the use of adjusted p-values due to the fact they are carrying out multiple comparisons. See details and other comments in the attached file.

Author Response

Response to Reviewer 1 Comments

First of all, we would like to thank the reviewers for their valuable comments, feedback and useful recommendations.

Their time taken to review our manuscript is much appreciated and helped to improve its quality. We have

taken their views onboard and have made the following changes. Below is the point-by-point response to

the comments and feedback of the Reviewer 1.

Point 1: Include a brief discusión/comments about the choice of the COI (cut-off -interval) in IgM/IgG. added

Response 1: A brief information about COI was added to lines 86-91 and 136-137.

Point 2: Take into account the problem with p-values when multiple comparisons are carried out. Methods of adjusted p-values, such as Benjamini-Hochberg, FDR, or similar should be applied. I prevent the authors of using Bonferroni, because this method is too strict. Some of the previously mentioned, I think could be appropriate.

Response 2: Bonferroni correction of p-values was applied, since this comment was also addressed by another reviewer, who strongly suggested to use this method. Description was added to lines 138-141.

Point3: DefineMAFinline140.

Response 3: MAF was defined in line 168.

Point 4: Check the head of manuscript. I think that ORCID is not properly incorporated in the manuscript.

Response 4: Thank you for the notification. ORCID number was removed from the head of the manuscript.

Reviewer 2 Report

The authors have carried out genetic association study of vitamin D metabolism pathway single nucleotide polymorphisms with susceptibility to asymptomatic COVID-19 in Kazakhstan. The study found G allele of rs731236 (VDR (TaqI)) and dominant mutation in rs10877012 (CYP27B1) as a risk factor for asymptomatic COVID-19. Though the study seems interesting, the study lacks certain crucial points. The below are some comments and suggestions to be considered by the authors.

1.      The title of the manuscript must be modified on the basis of the significant findings of the study.

2.      The sample size is very small [Only 134 cases and 46 controls]. Authors must provide the justification for the sample size for both the patients’ groups. This must be given in the materials and methods section.

3.      Why did authors not carry out Bonferroni’s correction for multiple testing? Since the study involved multiple SNPs, they must apply Bonferroni’s correction and reanalyze the results.

4.      Authors must mention the PCR conditions used for the SNPs, in methods section.

5.      Did the cases had any co-morbid condition i.e. diabetes etc.?

6.      Authors must do the age and gender based analyses for the SNPs.

7.      Authors must discuss about the rationale of the selection of SNPs, in introduction section.

8.      Authors must carry out LD and haplotype analyses and correlate the haplotype with the disease susceptibility.

9.      Authors have poorly presented the results section. Results section must be sub-divided according to each SNP analyzed. Gender and age based analyses must be presented as sub-section in the results.

10.  Material and methods: Authors mention that “Only participants with negative PCR were included in the study” ; In Table 1 they mention about case and control. Authors have not described clearly about case and control. They must mention the exclusion and inclusion criteria of Case and control groups. Moreover, they must mention these in the title ‘Study population/subjects’

11.  Authors must follow HUGO nomenclature for writing the human genes. The gene names must be in italics throughout the manuscript.

12.  The manuscript contains several grammatical and sentence formation errors which must be corrected.

Author Response

Response to Reviewer 2 Comments

First of all, we would like to thank the reviewers for their valuable comments, feedback and useful recommendations.

Their time taken to review our manuscript is much appreciated and helped to improve its quality. We have

taken their views onboard and have made the following changes. Below is the point-by-point response to

the comments and feedback of the Reviewer 2.

Point 1: The title of the manuscript must be modified on the basis of the significant findings of the study. 

Response 1:      The title was modified to “Association of CYP24A1 gene rs6127099 (A>T) polymorphism with lower risk to COVID-19 infection in Kazakhstan.”.

Point 2: The sample size is very small [Only 134 cases and 46 controls]. Authors must provide the justification for the sample size for both the patients’ groups. This must be given in the materials and methods section.

Response 2:      Since the data was collected during delta strain circulation and vaccination in Kazakhstan already started, it was complicated to find participants who have never had COVID-19 (self-reported), PCR negative and not vaccinated. That is why the research group was restricted to the limited number of samples. Moreover, initially, samples were collected for longitudinal study, but it appeared that substantial number of participants (n=134) had elevated IgM/IgG total antibodies.

Point3: Why did authors not carry out Bonferroni’s correction for multiple testing? Since the study involved multiple SNPs, they must apply Bonferroni’s correction and reanalyze the results.

Response 3:      Bonferroni correction of p-value was done. Added to lines 137-140. Since results section also faced consequent changes, edits were introduced to lines 174-178 and 179-187.

Point 4: Authors must mention the PCR conditions used for the SNPs, in methods section.

Response 4:      PCR conditions were added to lines 117-120.

Point 5: Did the cases had any co-morbid condition i.e. diabetes etc.?

Response 5:      Yes, they are indicated in Table 2 and there is no statistically significant difference between cases and controls.

Point 6: Authors must do the age and gender based analyses for the SNPs.

Response 6:      Table 3 covers adjusted odds-ratios (AOR) for SNPs based on modes of inheritance and adjusted for age, male gender and Kazakh ethnicity. Stratified analyses of SNPs based on age and gender is attached as Supplementary Table S1.

Point 7: Authors must discuss about the rationale of the selection of SNPs, in introduction section.

Response 7:      Elaboration of importance of selceted SNPs was added to introduction part to lines 51-71. New references from #13 to #20 were added.

Point 8: Authors must carry out LD and haplotype analyses and correlate the haplotype with the disease susceptibility.

Response 8:      Added to lines 126 and 133. New section 3.3. was introduced to results part of the manuscript lines 205-221. New Table 4 was added to line 223.

Point 9: Authors have poorly presented the results section. Results section must be sub-divided according to each SNP analyzed. Gender and age based analyses must be presented as sub-section in the results.

Response 9: Table 3 was modified and adjusted odds-ratios (AOR) were added to the table. Lines 197-201 were edited. Lines 202-204 were added. Lines 20-24 and 231-236 were edited.

Point 10:                      Material and methods: Authors mention that “Only participants with negative PCR were included in the study” ; In Table 1 they mention about case and control. Authors have not described clearly about case and control. They must mention the exclusion and inclusion criteria of Case and control groups. Moreover, they must mention these in the title ‘Study population/subjects’

Response 10:    Description of COI and definition cases and controls are added to lines 85-90 and 135-136.

Point 11:                       Authors must follow HUGO nomenclature for writing the human genes. The gene names must be in italics throughout the manuscript.

Response 11:    Thank you for the notification. Genes were changes to italics throughout the manuscript. They left as they are in the context of proteins.

Point 12:                      The manuscript contains several grammatical and sentence formation errors which must be corrected.

Response 12:    Proofreading was conducted.

Reviewer 3 Report

The article presents a simple but elegant proposal for understanding COVID-19 according to the genetic background of the human host with regard to the metabolic pathway of vitamin D, a paradigm so discussed in studies of the pathogenesis of COVID-19. I point out below, suggestions for improving the presentation of the text.

Table 1 - The VDR gene refers to the vitamin D receptor. VDR contains a different set of genetic polymorphisms, including   ApaI (rs7975232), BsmI (rs1544410), TaqI (rs731236), and FokI (rs2228570). The formatting of the table did not seem to make this information clear. I suggest including a space between this gene and the others for emphasis.
Do the numbers for HWE indicate the p-values?

In line 122, page 4 “185 persons accepted”. Does the use of the word “persons”  denote that you refer to distinct ethnic groups (for example, within the same region)? Otherwise, I suggest reviewing the writing of the terms in English.

In Table 3, what test for means was used? Were normality assumptions evaluated? I also suggest reviewing the formatting of this table. For some variables, the authors present only one line. could maintain the same pattern, and point out the statistical tests used.

For Figure 1, I strongly suggest separating the polymorphisms by function groups, and highlighting them in the graph. For example: Vitamin D receptor, Vitamin D binding protein, etc. This would certainly help the reader to understand which vitamin D metabolism pathway genes polymorphisms have the most impact on the pathogenesis of COVID-19.

Author Response

Response to Reviewer 3 Comments

First of all, we would like to thank the reviewers for their valuable comments, feedback and useful recommendations.

Their time taken to review our manuscript is much appreciated and helped to improve its quality. We have

taken their views onboard and have made the following changes. Below is the point-by-point response to

the comments and feedback of the Reviewer 3.

Point 1: Table 1 - The VDR gene refers to the vitamin D receptor. VDR contains a different set of genetic polymorphisms, including   ApaI (rs7975232), BsmI (rs1544410), TaqI (rs731236), and FokI (rs2228570). The formatting of the table did not seem to make this information clear. I suggest including a space between this gene and the others for emphasis.

Do the numbers for HWE indicate the p-values?

Response 1:      Table 1 was modified in accordance with suggestions.

Point 2: In line 122, page 4 “185 persons accepted”. Does the use of the word “persons”  denote that you refer to distinct ethnic groups (for example, within the same region)? Otherwise, I suggest reviewing the writing of the terms in English.

Response 2:      Line 149 was edited: “185 participants were recruited for this study”

Point3: In Table 3, what test for means was used? Were normality assumptions evaluated? I also suggest reviewing the formatting of this table. For some variables, the authors present only one line. could maintain the same pattern, and point out the statistical tests used.

Response 3:.     Genotypes are categorical variables, thus normality was not checked. Nevertheless, descriptive statistics included nonparametric methods due to limited sample size. Table 3 was modified and adjusted odds-ratios (AOR) were added with indication of p-values.

Point 4: For Figure 1, I strongly suggest separating the polymorphisms by function groups, and highlighting them in the graph. For example: Vitamin D receptor, Vitamin D binding protein, etc. This would certainly help the reader to understand which vitamin D metabolism pathway genes polymorphisms have the most impact on the pathogenesis of COVID-19.

Response 4:      Figure 1 was modified. Namely, SNPs were sorted based on in which gene they are located and gene names were included. Also, significance level p<0.001 was introduced to indicate strong difference between cases and controls in rs6127099. Not to overwhelm the figure, description of vitamin D metabolising fuctions were not added, since they are indicated in Table 1.

Round 2

Reviewer 2 Report

Authors have not revised the manuscript substantially. The major concern is still the very less sample size for controls. Authors must calculate the sample size and accordingly justification for sample size is given. However, if control sample size is not increased, the study will be under powered and such results cannot be considered for the conclusions drawn.
